# OpenReview forum: "Why Are Web AI Agents More Vulnerable Than Standalone LLMs? A Security Analysis"
_ICLR.cc/2025/Workshop/BuildingTrust — BuildingTrust_

### Official Review · Reviewer_WVKn · 2025-03-02
**Evaluating Web AI Agent Vulnerabilities – Valuable but with Some Limitations**

**Rating:** 7
**Confidence:** 3

**Review:**

Overview:
The paper presents valuable insights into the security risks of Web AI agents and introduces a novel fine-grained evaluation method. However, for a workshop submission, expanding the experiments, testing across different models, and refining the visual presentation would still be necessary.

Pros:
- Identifies key vulnerabilities in Web AI agents, providing meaningful security insights.
- Introduces a fine-grained harmfulness evaluation framework, improving jailbreak assessment.
- Well-structured ablation studies.

Cons:
- Limited Scale of Experiments: The evaluation primarily relies on one model (GPT-4o-2024-0806). Given the diversity of Web AI agents, evaluating on multiple models (e.g., Claude, Gemini, open-source models) would strengthen the claims. The number of test cases appears to be limited (10 harmful requests across different categories). A larger dataset would provide more robust results.
- Lack of Clarity in Figure 2: Figure 2 plays a crucial role in illustrating the differences between Web AI agents and standalone LLMs, but it lacks clarity and a professional layout. A more visually structured version with clearer labels and explanations would improve readability.
- Unclear Definition of "Real Websites": The authors mention testing on "real websites" but do not clearly define what those websites are or how they were selected. It is also unclear whether the AI agent was fully interacting with the real web or whether interactions were limited in some way (e.g., sandboxed environments).

---

### Official Review · Reviewer_8u3R · 2025-03-02
**This paper examines why web-AI agents are more vulnerable to malicious prompts than standalone LLMs, offering an insightful evaluation framework with empirical evidence.**

**Rating:** 7
**Confidence:** 3

**Review:**

As the demand for AI agents rises, it is essential to understand why AI agents are susceptible to malicious user prompts. This paper addresses this issue by investigating why web-AI agents are more susceptible to following malicious prompts than their safety-aligned standalone LLMs.

I believe the paper has done a very good job of assessing the key factors contributing to fragility and has presented a fine-grained evaluation framework with empirical evidence.

I think the paper would have been stronger with a larger sample size and by using a more diverse set of LLMs, rather than just GPT-4.

This paper aligns with the workshop objectives and will clearly benefit the research community.

---

### Official Review · Reviewer_x3Us · 2025-03-02
**Interesting paper of value to the community but evaluations can be much more comprehensive**

**Rating:** 6
**Confidence:** 4

**Review:**

This paper prompts web agents from the OpenHands framework to carry out malicious requests and observes differences in jailbreaking success rate of a standalone LLM versus one in the agentic framework.

Pros:
+ interesting study with a well-designed set of ablations for this particular agentic framework

Cons:
- The authors should baseline with a stronger jailbreaking dataset. GPT-4o is far from being a model that achieves 0% jailbreaking success rate. A quick Google search for "gpt-4o jailbreaks" identifies multiple claimed successful jailbreaks and there are works like https://arxiv.org/abs/2404.02151 that even claim a 100% success rate.
- The authors should include other agentic frameworks and add ablations for those other possible system design choices. For example, Anthropic provides a reference implementation for their computer use agent that is much simpler and does not include obviously bad design choices like embedding user requests in the system prompt: https://github.com/anthropics/anthropic-quickstarts/tree/main/computer-use-demo Another popular web agentic framework to include is VisualWebArena: https://arxiv.org/abs/2401.13649 WindowsArena also provides a popular scaffolding: https://arxiv.org/abs/2409.08264v1 Finally, official implementations from OpenAI like Operator (https://openai.com/index/introducing-operator/) would be interesting to test once they are broadly available.
- The authors should include more and different backbone models instead of only GPT-4o.

---

### Decision · Program_Chairs · 2025-03-04

Accept